# Aerial Transmission of the SARS-CoV-2 Virus through Environmental E-Cigarette Aerosols: Implications for Public Policies

**DOI:** 10.3390/ijerph18041437

**Published:** 2021-02-03

**Authors:** Roberto A. Sussman, Eliana Golberstein, Riccardo Polosa

**Affiliations:** 1Institute of Nuclear Sciences, National Autonomous University of Mexico, 04510 Mexico City, Mexico; 2Myriad Pharmaceuticals Ltd., 1010 Auckland, New Zealand; eliana.golberstein@myriad.nz; 3Center of Excellence for the Acceleration of Harm Reduction (CoEHAR), University of Catania, 95123 Catania, Italy; polosa@unict.it

**Keywords:** SARS-CoV-2, COVID19, vaping, smoking, facemasks, risk analysis

## Abstract

We discuss the implications of possible contagion of COVID-19 through e-cigarette aerosol (ECA) for prevention and mitigation strategies during the current pandemic. This is a relevant issue when millions of vapers (and smokers) must remain under indoor confinement and/or share public outdoor spaces with non-users. The fact that the respiratory flow associated with vaping is visible (as opposed to other respiratory activities) clearly delineates a safety distance of 1–2 m along the exhaled jet to prevent direct exposure. Vaping is a relatively infrequent and intermittent respiratory activity for which we infer a mean emission rate of 79.82 droplets per puff (6–200, standard deviation 74.66) comparable to mouth breathing, it adds into shared indoor spaces (home and restaurant scenarios) a 1% extra risk of indirect COVID-19 contagion with respect to a “control case” of existing unavoidable risk from continuous breathing. As a comparative reference, this added relative risk increases to 44–176% for speaking 6–24 min per hour and 260% for coughing every 2 min. Mechanical ventilation decreases absolute emission levels but keeps the same relative risks. As long as direct exposure to the visible exhaled jet is avoided, wearing of face masks effectively protects bystanders and keeps risk estimates very low. As a consequence, protection from possible COVID-19 contagion through vaping emissions does not require extra interventions besides the standard recommendations to the general population: keeping a social separation distance of 2 m and wearing of face masks.

## 1. Introduction

The current COVID19 pandemic has intensified scientific interest in aerial pathogen transmission through bioaerosols, which are classified conventionally as “droplets” or “aerosols” for aqueous respiratory droplets with diameters above and below a 5 mm (micrometers) cut off. Direct transmission of the SARS-CoV-2 virus through “droplets” across short distances is a fact acknowledged by the public health community and ratified by the WHO [1] and the CDC [2]. Indirect transmission across larger distances by “aerosols” has also been observed (especially in hospital wards [3]), but its scope and frequency remain controversial [4,5,6]. However, we shall avoid this excessively simplified (and merely conventional) binary classification into “droplets” vs. “aerosols”, with the term droplets (without quotation marks) denoting generic respiratory droplets of any given diameter.

There is a comprehensive scientific literature on aerial pathogen transmission, mechanisms of respiratory droplet generation, viral transport and dynamics of respiratory droplets emitted by different respiratory activities, such as respiration [7], vocalization [8], coughing [9] and sneezing [10]. However, there is no empiric evidence of aerial transmission of the SARS-CoV-2 virus (or any pathogen) through environmental e-cigarette aerosol (ECA) or environmental tobacco smoke (ETS) exhaled by infected vapers or smokers. In order to address this lack of evidence and proper elaborate research, a comprehensive study [11] was undertaken to infer and assess rigorously and extensively the plausibility, scope and risks of pathogen transmission through exhaled ECA (previous literature consisted of only three opinion pieces presenting weak arguments [12,13,14]). In the present paper we consider and extend the findings of [11] in order to contribute to the setting up of guidelines for public policies on vaping and smoking in the context of containment, prevention and mitigation strategies in the COVID-19 pandemic. We believe this is a relevant mission, given the fact that these policies affect millions of vapers and smokers (and those around them), who need to share indoor and outdoor spaces at varying levels of home confinement and mobility constraints, including compulsory domestic confinement in the form of complete or partial lockdowns, closure of non-essential enterprises, restaurants, colleges, leisure activities, mass gatherings.

The global response to the pandemic has produced the loss of millions of jobs [15] massive confinement in many jurisdictions, but conditions have varied in time and in the geography. Some economies have partly reopened, at least temporarily, when containment measures seem to have reduced contagion rates, permitting relaxation of involuntary confinement, but most jurisdictions retain different degrees of economic activity limits with varying social contact limitations (for a review of global public policies see [16]).

The social effects and rapid changes associated with the COVID-19 pandemic have produced specific psychosocial problems that have impacted negatively on the mental health of the population living under these strict steps. Forced isolation has contributed to anxiety and depression [17], factors which should probably lead to a rise in the intake of psychoactive stimulants, alcohol, cigarettes and nicotine products in certain individuals to alleviate tension and negative feelings. These situations provide an appropriate framework to understand the need for evidence-based arguments to address relevant issues on smoking and vaping in the context of the COVID-19 pandemic.

Concern on the possible virus transmission through exhaled ECA or ETS is perfectly legitimate but needs to be placed in its proper context, specifically in reference to transmission through other respiratory activities within prevention and containment measures to address the COVID-19 pandemic. Our aim is to contribute useful knowledge that can aid health authorities and those responsible for public policy planning to better understand how to improve the life and welfare of millions of vapers and smokers (and their families) currently under this global pandemic hardship. Regrettably, this legitimate concern has prompted some public health authorities to overreact by enacting smoking and vaping bans in outdoor public open spaces, as for example in Spain [18,19]. This is an overzealous and invasive protective measure that lacks evidence or scientific justification and whose public health benefit is doubtful.

The following disclaimers must be issued: this article does not deal with health risks or potential hazards resulting from exposure to ECA that are not directly linked to COVID-19, with emphasis on its potential contagion through respiratory droplets transported by its exhalations. While our main interest is focused on the effects of possible SARS-CoV-2 transmission through exhaled ECA, under certain nuances some aspects of our assessments and discussion apply to ETS (see Section 6).

## 2. Smoking and Vaping as Risk Factors for COVID-19

Although the main topic of this paper is to examine the possibility and scope of COVID-19 contagion through exhaled ECA and ETS, it is necessary to comment on the legitimate questions about the possible association between smoking and vaping vs. infection among vapers and smokers and the various stages of COVID-19 related disease. it is also necessary to review the literature investigating how smokers and vapers cope under the specific conditions of the pandemic.

The WHO [20] and several studies [21,22,23] have identified smoking as a risk factor for COVID-19. This is a rational assumption, as smoking is a major factor leading to reported vulnerability conditions for COVID-19, such as cardiovascular ailments, diabetes or chronic lung disease [24,25]. While several studies [26,27,28] have shown a significant under-representation of smokers among subjects diagnosed with COVID-19 admitted to hospitals, a systematic meta-analysis [29] has reported that few smokers are actually admitted to hospitals, but that once hospitalized they face a higher risk for severe outcomes than non-smokers. These findings have prompted research [30,31] to explore the possibility that nicotine may provide a protective effect by interfering with the biochemistry of viral infection or the deadly overreaction of the immune system, since more severe outcomes of hospitalized smokers might be consistent with their sudden termination of nicotine consumption once admitted to hospitals. However, there is also skepticism on the outcomes of studies showing underrepresentation of smokers among hospitalized patients [32,33], while the nicotine protective hypothesis remains so far untested. Therefore, the interrelation between smoking, nicotine and COVID-19 remains so far inconclusive.

Several sources have argued that vaping is also a risk factor for COVID-19 [34,35,36], mostly on the basis of very indirect evidence of lung inflammatory processes reported from in vitro research, animal models and physiological harms detected in pulmonary tissue extracted from small samples of human vapers, all of whom are former or current smokers (see [37] for a review of these studies and [38] for a critical appraisal). To claim that these findings provide strong evidence that vaping is a risk factor for COVID-19 seems to be inconsistent with the apparent absence of vapers among registries of hospitalized or seriously ill COVID-19 patients. In fact, as compared to reports of smoking and other comorbidities, vaping habits among hospitalized COVID-19 patients have not been collected up to this date in epidemiological studies.

A recent study by the University of Stanford [39], based on a self-reported internet survey collecting data among young people aged 13 to 24 years up to 14 May 2020, reported that ever e-cigarette use (exclusive and dual use of tobacco cigarettes) increases five-fold their odds of a positive COVID-19 result in a PCR test with respect to never users. However, as a contrast, actual vapers have the same odds for a positive COVID-19 result as never users (thus suggesting lack of biological plausibility). Also, the extrapolation of the surveyed sample to the USA population weighed by the 2018 census is inconsistent with the number of tests performed in this age group at the time of the study (see [40] for criticism of this article and the authors’ response).

There are very few studies on smoking/vaping consumer habits during the COVID-19 pandemic. The associations between vaping and self-reported diagnosed/suspected COVID-19 was examined recently by a research team from University College London [41], based on cross-sectional data from the longitudinal online study of UK adults: the **HE**alth **BE**haviour during the **CO**VID-19 pandemic (HEBECO) study. They found no association between diagnosed/suspected COVID-19 among never, current, and ex-vapers. Other findings are

Among recent ex-vapers, 17.4% quit vaping as precaution for COVID-19, but 40.7% considered taking up vaping again since COVID-19, mostly out of stress and boredom.Among current vapers: 50% did not change vaping habits, 40% increased consumption and 10% decreased consumption.

Another study on smoking and vaping habits under COVID-19 was conducted by an Italian team [42]. The study is based on a self-reported internet questionnaire on a sample of 1925 participants: exclusive cigarette smokers, dual users of cigarette and e-cigarettes, dual users of cigarette and heated tobacco products, former smokers, exclusive users of e-cigarette, exclusive users of heated tobacco products and never smokers. The main findings are:Dual users of cigarette and e-cigarette and exclusive cigarette smokers perceived that their daily consumption has slightly decreased.Most exclusive cigarette smokers have considered quitting, but most exclusive e-cigarette users have not considered stopping the use of e-cigarettes.About one third of former smokers declared thoughts about starting to smoke again.

In spite of their limitations (being cross sectional self-reported surveys), these studies illustrate how COVID-19 may contribute to the reinforcement of various intentions or behavioral trends. Many smokers continued to smoke, despite being aware of the harms from smoking (even ex-smokers declared intentions to smoke again). A continuous stream of troubling media reports and attempts by health authorities to discourage the use of e-cigarettes described as a COVID-19 risk factor [43], persuaded some vapers to consider to quit vaping or to decrease consumption, but a substantial number of vapers kept vaping and even increased consumption. These behaviors and behavioral patterns are likely to reflect a balance between fears of infection or serious illness (often fed by the media or health authorities [43]) and the need to cope with cravings and tension in pandemic circumstances. These are significant issues that require further research.

## 3. Exhaled ECA as a Visible Respiratory Flow: Direct Exposure

The physical and chemical properties of exhaled ECA are essential to infer its capacity to transmit the SARS-CoV-2 virus through the transport of respiratory droplets. We include here a review of these issues. Readers interested in technical details are advised to consult [11] and references cited therein.

Since about 90% of inhaled ECA is retained by the respiratory system [44], ECA is a strongly air diluted aerosol, whose particulate phase is made of submicron liquid droplets (i.e., diameters below 1 mm) composed of propylene glycol (PG), vegetable glycerin (VG), nicotine and water [45], with similar composition for its gas phase. As opposed to the “airborne” pathogen transmission for other respiratory activities, vaping involves an “ECA-borne” transmission carried by a different fluid in which respiratory droplets would be accompanied by a far larger number of ECA droplets (bioaerosols particle numbers are in general far fewer than in non-biological aerosols [46,47]).

The diameter distributions of ECA droplets peak at submicron values [11] This should also hold for respiratory droplets that would be transported by ECA (see Section 4). The few larger ECA and respiratory droplets leave the flow of the carrier fluid at a rate that depends on their size and follow ballistic trajectories. The larger ones (typical diameters over 20 mm) rapidly settle on the ground or deposit on surfaces before evaporation, while intermediate ones (typical diameters 2–20 mm) might evaporate before settling and remain buoyant longer times. Having little inertia, submicron droplets follow the flow of the carrier fluid, which for the involved distance scales and temperature gradients can be considered to a good approximation as isothermal. Optical properties of liquid droplets in large numbers (light scattering, see [11]) make the flow of ECA a visible cloud [46,47], i.e., the droplets act as visual tracers of the associated respiratory flow. In fact, aerosols with submicron droplets (like ECA) approximately evolve like gases with its particles behaving as molecular contaminants and are thus widely used to visualize respiratory flows [48] (even tobacco smoke has been used for this purpose [49]).

The fact that exhaled ECA offers an effective visualization of the expired flow is a very significant property that distinguishes vaping (and smoking) from other respiratory activities potentially transmitting pathogens. This property has an important psychological dimension: bystanders seeing the expiratory flow potentially carrying pathogens can instinctively (without scientific training and without undertaking laboratory experiments or computations) position themselves to avoid direct exposure, something impossible or very hard to do with other expirations that are invisible. This is also relevant for safety and precautionary concerns, as visualization makes it absolutely clear that direct exposure risk distances are in the range 1–2 m but only in the direction of the exhaled jet, with individuals placed in other directions only facing an indirect exposure risk (whether they wear face masks or not). Nevertheless, it is prudent to maintain a 2 m separation distance from everyone vaping if not wearing a face mask.

The instinctive appreciation of the above-mentioned direct exposure distance and direction was rigorously corroborated in [11] by modeling exhaled ECA as an intermittent turbulent jet, made of ECA diluted in air, evolving into an unstable puff. As long as the exhalation lasts the jet trusts ECA and its accompanying respiratory droplets (which are also submicron, see Section 4) in the direction of the exhaled jet. From estimated exhalation velocities between 0.3 m/s and 3 m/s and assuming a horizontal exhalation, the model predicts a distance reach for the exhaled jet/puff system between 0.5 to 2 m.

The dynamical parameters inferred above assume the low intensity ‘mouth to lung’ (MTL) vaping style practiced by the vast majority (80–90%) of vapers, involving a mouth hold before lung inhalation and using low powered devices (either starter kits, closed systems or pods). However, 10–20% of vapers practice the more intense ‘direct to lung’ style using high powered tank devices. As show in [11] this style involves larger exhalation velocities and distance spreads of over 2 m. In this paper we will only consider the MTL style, as it is the most representative of vapers worldwide.

Another factor that needs to be considered is the potential effects on respiratory droplets due to the bactericidal and virucidal properties of glycols contained in ECA, such as PG and VG, which have been tested experimentally. A review of the data (see [11] reported in these experiments indicates that environmental disinfection by these glycols is unlikely to occur under typical e-cigarette use conditions. While there is no experimental evidence that disinfection by these glycols would work on the SARS-CoV-2 virus, there is also no evidence nor theoretical reasons to assume that these compounds (or other compounds present in ECA) could somehow enhance its infective action. Nonetheless, suitable studies should be established to assess these possibilities even outside the context of vaping.

## 4. Indirect Exposure in Shared Indoor Spaces

Once the fluid injection terminates (exhalation ends) the ECA jet becomes a highly turbulent roughly ellipsoidal puff that is rapidly disrupted by turbulent mixing from entrained surrounding air, with the trusted ECA and respiratory droplets drifting into the surroundings, carried by indoor air currents and remaining buoyant for long times (hours), thus leading to indirect exposure. To estimate indirect exposure through droplet dynamics it is necessary to incorporate into the model the effects of turbulent air mixing and thermal convection, as well as (ideally) more realistic conditions, such as a ventilation regime, heat emission from people and furniture and moving sources, all of which requires more advanced theoretical modeling and computational methods of fluid mechanics (as for example in [50]). A more complete study would also have to consider environmental effects (especially temperature and relative humidity) on the dispersing droplets and even the viral particles themselves (see a review on available evidence on the SARS-CoV-2 virus [51,52]). Instead, we examined indirect exposure in shared indoor spaces, assuming uniformly spread droplets, through a simplified exponential risk model based on the rates of expired viral load through various respiratory activities of actual SARS-CoV-2 data.

### 4.1. Respiratory Droplets That Should Be Carried by Vaping

To infer and evaluate indirect exposure risk from expiratory activities we need observational data on their expired volume, rates of respiratory droplet emission and droplet diameters. This data exists for breathing, vocalizing, coughing and sneezing, but not for vaping and smoking. Given the lack of experimental evidence on these parameters for exhaled ECA, we need to resort to appropriate respiratory proxies that resemble vaping and on which such evidence exists. To accomplish this task we undertake the following steps (see [10] for details):(1)We examine the data on respiratory mechanics of cigarette smoking as a proxy to infer and estimate the exhaled volume and other respiratory parameters of vaping. This is justified, as most vapers are ex-smokers or current smokers,(2)Since vaping involves mouth inhalation by suction through a mouthpiece, we review the available literature on the effects of the inspiration/expiration routes and of mouthpieces and nose clips on respiratory mechanics.(3)Considering the data from the previous steps, we estimate the exhalation velocities associated with vaping and notice that they are comparable to measured velocities of mouth breathing. This suggests that mouth breathing can be considered as an appropriate proxy to estimate droplet emission from vaping.

The data from cigarette smoking and mouth breathing gathered by these steps suggests that vaping should:release on average a tidal volume of 700–900 cm^3^ exhaled ECA diluted in air,produce low emission rates: 6–200 (mean 79.82, standard deviation 74.66) respiratory droplets per puff, overwhelmingly in the submicron range (hence, they should be really droplet nuclei as droplets of this size evaporate instantaneously once exhaled).

Submicron respiratory droplet nuclei possibly transported by ECA fall in the range of diameters denoted in medical literature as “aerosols”. There are claims that these small droplets might play an important (so far unaccounted) role in spreading the SARS-CoV-2 virus [53,54], as there is evidence that this spread has occurred [3] and it is known (from droplet dynamics) that they remain buoyant, either as droplets before evaporation or as nuclei, for long periods (hours) and drift long distances (meters). These claims could be further supported by the detection of SARS-CoV-2 viral RNA in ventilation systems of hospital rooms [55], as well as by experimental evidence showing that SARS-CoV-2 virus in aerosol droplets remains viable and stable for 3 h [51,52,56]. However, these were highly idealized experiments in which the artificially generated bioaerosol might not be an accurate simulation of droplets (especially small ones or their nuclei) generated in the respiratory system. Their airborne evolution in closed chambers might be unrepresentative of realistic conditions in indoor and outdoor environments. Also, detected RNA of SARS-CoV-2 does not necessarily indicate the presence of a viable infectious virus [57]. Therefore, the scope and frequency of COVID-19 contagion from this type of droplets (which would include the type transported by ECA) remain controversial (see [6,7]).

### 4.2. Relative Risk Model

Given the parameters inferred above, we assessed in [11] the risk of SARS-CoV-2 contagion through indirect exposure to respiratory droplets in shared indoor spaces by means of a simplified adaptation (which incorporates vaping) of the exponential dose-response reaction model of Buonanno, Morawska and Stabile, henceforth BMS [58] (see also [59]). BMS base their analysis on the notion of an infective quanta: the droplet dose necessary to infect 63% of exposed individuals, with the basic quantity defined as the rate of emitted quanta per hour ER_q_, proportional to the viral load (RNA copies per mL) taken from collected data on the SARS-CoV-2 virus, the total volume of exhaled droplets (in mL), the breathing frequency and the exhalation rate. To evaluate ER_q_ for different expiratory activities, MBS use available observational data on emission rates and droplet diameters for breathing and speaking, which we adapt for vaping.

However, ER_q_ also depends on the duration of the expiratory activity. Breathing involves low amounts of emitted quanta, but it carries on continuously and is not suspended while people talk or cough, and also when they vape or smoke. Talking and coughing emit significantly higher values of ER_q_ than breathing, but are of short duration, while vaping is also of short intermittent duration and emits just slightly higher ER_q_ than breathing (but very close). Typically, vaping involves 160–200 puffs per day (in a 16 h journey), which means 2 min employed in 10–13 breaths per hour among the roughly total average 600–1400 breaths per hour for average adults in rest breathing.

As a consequence of its low intensity and intermittent nature (each puff is roughly one breath long), vaping adds every hour just a minuscule (roughly 1%) increase of emitted quanta on top of those quanta emitted by continuous (unavoidable) rest breathing, which can be considered as the baseline “control” state. As a reference, normal speech for 6 min in one hour adds roughly 44% extra infective quanta over this control state.

BMS use an analytic expression for the exponential risk model and consider probability distributions and Monte Carlo simulations to account for individual variability of infective parameters and susceptible individuals. Instead, we define in [11] a relative risk of indirect exposure with respect to the above-mentioned control state as the quotient of ER_q_ associated with a given expiratory activity with respect to ER_q_ for breathing, considering for every expiratory activity (speaking, coughing and vaping) the fraction of breaths per the hour it lasts. We also simplify the model of BMS by considering only median values (50% percentiles) of their probability distributions. Under these assumptions, our quotient that defines the relative exposure risk provides a good approximation to the analytic expression of BMS and to the risk model of their earlier paper [59].

Assuming that the submicron respiratory droplets from vaping (and other expirations) have been spread uniformly through an indoor space and considering recent data used by BMS on SARS-CoV-2 viral load and other infection parameters, as well as their data on droplet size and emission rates and our adaptation of this data to vaping, we evaluated in [11] these relative risks for a home and restaurant scenarios (12 and 3 h total exposure) with natural and mechanical ventilation. The resulting values of added risks computed with respect to the control case are:1% for vaping (160 daily puffs, 16 h journey)44% for continuous speaking 10% of time (6 min every hour), up to 176% for speaking 40% of time (24 min every hour)over 260% for coughing 30 times per hour.

Notice that these are relative risks with respect to a control state defined by continuous breathing without vaping, speaking or coughing. As a consequence, these results hold for both scenarios and ventilation regimes, though the absolute number of emitted quanta vary significantly depending on the exposure time, volume of indoor space, number of susceptible individuals and type of ventilation regime (natural and mechanical). We find that mechanical ventilation decreases absolute risk for indirect exposure by an order of magnitude for each activity. We display in Figure 1 a sketch of the area of direct exposure for breathing, vaping, speaking and coughing, together with the relative risks for indirect exposure with respect to the control state of continuous breathing (computed in our risk model). Notice that only vaping allows for the direct visibility of the area of direct exposure.

## 5. Face Masks

We did not assume universal wearing of face masks in the analysis of Section 4. While this assumption is well justified in a home in family scenario in which masks are rarely worn, it is important to discuss its implications in scenarios where universal wearing of face masks is recommended and complied with. Nevertheless, it is important to remark that even at the lower level of protection without face masks the analysis of [11] shows that vaping in a shared indoor space adds only a minuscule additional risk (1%) to indirect exposure with respect to those risks already existing from continuous breathing. However, as we show in Section 5, bystanders wearing face masks should be reasonably well protected once placing themselves outside the area of direct exposure that (in the case of vaping) is clearly visible and delineated by the exhaled jet (see Figure 1).

Since vaping requires an (at least temporary) removal of the vaper’s face mask, there could be concerns that in shared spaces where universal face mask wearing is recommended the exposure to vaping exhalations could necessarily represent a significant (and worrying) increase of contagion risk to bystanders, even if the latter wear face masks. The key argument behind these concerns is the notion that mask protection against pathogens carried by respiratory droplets is only effective when it is reciprocal (i.e., both the droplet emitter and receiver wear masks). Accordingly, vaping would not meet this protective criterion because of the removal of the face mask. This increase of risk to bystanders (even if wearing face masks) when there is no reciprocal masking has been widely conveyed in web pages and public messaging propagating preventive information and resources to address risks of COVID-19 contagion [60]. Concern for this risk is the main argument behind the outdoor ban on vaping and smoking by the Spanish authorities [18,19] As we argue below, this argumentation is unsustainable once we undertake a more realistic risk evaluation.

It is known that N95 respirators afford effective protection to wearers [61], but empiric evidence on mask protection from inward external emissions for the wearer (that would support the need for reciprocal masking) in the most commonly used surgical masks is scarce, as most studies on mask filtration efficiency deal with outward emission by masked subjects [62,63]. Inward penetration of a stream of virus transported through artificially generated aerosols into surgical masks worn by mannequins was examined in two experimental studies. In [64] penetration from the nebulizer stream into the mask worn by a mannequin located at close range was rather high (67%) compared with the much lesser 10% penetration into the N95 respirator. Virus penetration (in terms of virus titer associated with a 5 × 10^5^ FPU emission) into the masks worn by a receiving mannequin was measured in [65] (see its Figure 2) to be significantly higher (83% cotton mask, 53% surgical mask) when the emitting mannequin was not wearing a mask compared to 31% and 24% (cotton and surgical masks) when the spreader wore a surgical mask. A receptor wearing a fitted N95 respirator made a big difference: penetration was only 10% and 4% with the spreader being mask-less and when wearing a surgical mask. However, the separation distance is crucial: in [65] measurements were performed at 50 cm separation, with percentages roughly decreasing by one third when made at 1 m separation.

The experiments in [64,65] described above provide (in spite of their idealization) empiric support for the preference of reciprocal mask wearing at close range, especially for cotton and surgical masks. While not measured in [65], it is evident that droplet penetration would decrease even further at separation distances beyond 1 m. Thus, the available empiric evidence supporting a significant loss of protection to a receiving bystander by the lack of reciprocal mask wearing is strictly valid only when he/she is located in the area of direct exposure. As a consequence, the lack of reciprocal face mask wearing (by the need for face removal to vape) does not invalidate our risk analysis, which is explicitly valid for bystanders placing themselves away from the visible direct exposure range (<2 m within the exhaled jet), with all those located everywhere else in the vicinity subjected only to indirect exposure, a situation not contemplated in these experiments (and also not contemplated in info-graphs and announcements in media conveying information and resources as in [60]).

Direct exposure to pathogens associated with vaping would occur from a stream of droplets transported by the exhaled ECA jet, whereas indirect exposure is associated with submicron droplet nuclei that are rapidly dispersed by turbulent air currents after the exhaled jet transporting them evolves into an unstable turbulent puff. Wearing a mask is much more protective for bystanders under this indirect exposure to droplets (most of them submicron nuclei denoted usually as “aerosols”) dispersed in a much larger air volume, even if they can drift for extended times along erratic trajectories. The protection afforded by a face mask against exposure to these small droplet nuclei is necessarily much more efficient, and not comparable, to the protection afforded against direct exposure characterized by a rapidly moving directional stream of droplets localized in the much smaller volume of the exhaled jet (all this besides the fact that the scope of SARS-CoV-2 transmission through “aerosols” is still uncertain and controversial).

In fact, to know whether the emitter is masked or not loses relevance under the conditions of indirect exposure to submicron droplet nuclei (i.e., “aerosols”) in large spaces. While an emitter can be identified in a home scenario with few household family members, in larger micro-environments (restaurant terrace or outdoors) an emitter might be very hard (or impossible) to identify among the many occupants, so that the risk evaluation in practical terms rests mostly on the protective gear worn by the receiver.

If face masks are universally worn in a home scenario (an extremely unusual situation) or in any other shared indoor or outdoor space it would be necessary to recalculate the risk assessment undertaken in [11] since the baseline control state of continuous breathing would involve a lower level of emission of infective quanta (depending on the outward filtration efficiency of the masks). However, while the intermittent emission from a vaper not wearing a mask (while vaping) remains the same, the face masks worn by everybody else would protect them from indirect exposure to this emission. As we have argued, invoking a sharp increase of risk for a lack of reciprocal masking makes no sense in large open spaces and can be reasonably handled by recommending bystanders to wear face masks and avoid the area of direct exposure (see Figure 1). It can be argued that vapers could remain mask-free for periods longer than the duration of intermittent puffs when vaping, but this depends on the incentives that the social context to induce them (or for those who eat and drink) to remain mask-free. We discuss this issue in Section 7.2.

## 6. Vaping vs. Smoking

Like ECA, ETS (environmental tobacco smoke) is also an aerosol whose particulate matter lies overwhelmingly in the submicron range. However, its solid and liquid particles (the TAR: tobacco aerosol residue) and its gas phase are characterized by a considerably higher level of chemical complexity and toxicity than ECA gas phase and particles (droplets made of PG, VG, nicotine and water [45]). Unlike ECA, whose only source is the mainstream emission from the exhalation of the vaper, ETS has two sources: in addition to the mainstream emission from the exhalation of the smoker, approximately 80% of the aerosol mass emitted into the environment comes from the side stream emission from the cigarette’s burning/smouldering tip [66].

As far as the characteristics of potentially carried respiratory droplets, distance for direct exposure and indirect exposure risks to SARS-CoV-2, the results obtained in [11] (summarized in Section 3, Section 4 and Section 5) apply only to mainstream ETS, as side stream emissions do not come from the respiratory system. As a consequence, pathogen transmission (including SARS-CoV-2) is a truly minor issue among health hazards from indoor exposure to ETS.

We emphasize that mitigation and prevention policies must bear in mind that, aside for SARS-CoV-2 transmission, vaping and smoking in indoor spaces represent completely different exposure risks. Studies of exhaled ECA that express concern on health risks from exposure to its “particles” [67] or from their deposition in the respiratory system [68], often overlook the fact that these “particles” are liquid droplets made of low toxicity compounds: PG, VG, nicotine and water [45]. There is no evidence of harm to bystanders exposed to exhaled ECA derived from inhaling these droplets, which are not comparable with particulate matter of combustible sources like ETS or air pollution, even if their number densities and diameters might be comparable.

While ETS is a serious indoor pollutant (specially in poorly ventilated spaces), exhaled ECA poses negligible health risk to bystanders. This assessment follows not only from the much higher content of toxicants in the particulate and gas phases of ETS (especially side stream emissions), but from the duration of the exposure, a crucial factor that determines the total load of inhaled toxicants. Bystanders are exposed to exhaled ECA in indoor spaces for very short periods, as its mean life is 10–20 s per exhalation, while their exposure to ETS is of long duration with mean life up to 40 min per exhalation (see [69,70]). This significant difference follows from their distinct physicochemical properties: ECA droplets rapidly evaporate into the rapidly diluting and dispersing supersaturated gas phase. As a contrast, both phases of ETS have a large non-volatile content that does not evaporate, but ages and lingers long periods in the environment, with its solid and liquid particles slowly settling gravitationally or depositing in surfaces and walls [69,70].

## 7. Implications for Prevention and Containment Policies

### 7.1. Home Confinement

The home scenario is especially relevant to assess COVID-19 transmission from vaping and other expiratory activities during home confinement, which is the indoor scenario that has affected most of the population in the current pandemic at global scale. Home confinement is relevant, not only when containment measures have required a strict mandatory lockdown, but also under less strict conditions of a mitigating strategy which allows for a partial reopening of economic activity, but still advises the population to stay at home as long as possible.

The pandemic has been characterized by a broad geographical and temporal variance in the severity of conditions, with increasing rates of infection and hospitalization leading to restrictions on social and business practices, closure of restaurants, bars, shops and non-essential industries, both of which suggest a rise in the proportion of the population at least partially under home confinement. For example, this was reported in a survey conducted in an important jurisdiction like New York City between September and November across 46 thousand data points, showing that 73.84% of new COVID-19 cases come from in-home meetings, 7.81% from healthcare delivery and just 1.43% from bars and restaurants [71].

The home scenario fits the indoor conditions that large numbers of vapers and smokers (and their families) must endure for a range of large periods under home confinement in which face masks are not usually worn. The 2 m separation to avoid direct exposure and the risk assessment for indirect exposure, summarized in Section 3, Section 4 and Section 5, provide valuable contextual information for safety policies in this scenario (face mask wearing is not an issue as they are seldom worn at home). Vaping with the average frequency of 160–200 puffs in a 16 h journey only adds a minuscule (~1%) extra contagion risk by indirect exposure with respect to the control case scenario of continuous breathing. It is therefore crucial that preventive measures should take into account that recommending abstention from vaping at home merely produces a negligible improvement in protection, with the potentially undesired effect of increasing the level of stress and anxiety of vapers and their families under confinement. Containment and prevention strategies should also take into account that promoting abstinence from vaping at home makes no sense when speaking (whose abstinence is not advised by a sensible policy) exposes household members to a substantially greater increase in relative risk (44% to 176%for speaking 6 and 24 min every hour).

As we commented in Section 6, containment and prevention measures must distinguish between exhaled ECA and ETS. While exposure to ETS under home confinement can be hazardous pollutant for vulnerable individuals (specially in poorly ventilated spaces), it is not an important transmission vector for COVID-19.

### 7.2. Restaurants and Outdoor Environments

The prohibition of vaping inside homes has not been proposed by prevention or containment strategies (it would be an extremely ineffective and invasive action that would not increase safety), though many jurisdictions have banned vaping in publicly shared indoor spaces before the current pandemic: restaurants, bars, malls, bus and train terminals, airports, etc. However, vaping is typically tolerated outdoors and often in enclosed or open terraces adjacent to bars and restaurants, areas that may not be closed to the public under less extreme pandemic conditions.

As opposed to home scenarios where face masks are seldom worn, prevention and mitigation measures strongly encourage universal face mask wearing in all publicly shared indoor and outdoor spaces, a suggestion that is usually taken up by the public in most countries and regions with fair implementation. As we argued in Section 5, the increase of risk due to the temporary face mask removal needed to vape is practically inconsequential if bystanders wear face masks and avoid the area of direct exposure (the visible exhaled jet, see Figure 1). However, the frequency and consequences of face mask removal depend on the incentives for this in specific contexts of social interaction. We can broadly highlight the following representative contexts:*A bar or restaurant terrace where vaping is allowed*. While keeping the recommended 1.5–2.0 m separation distance, a convivial atmosphere provides a lot of incentives to remove face masks (necessary for eating and drinking). People eating and drinking tend to speak to each other. If pandemic containment conditions become sufficiently relaxed to tolerate the risk involved when mask-less patrons spray respiratory droplets while eating, drinking and speaking, then there is no reason to object to the convivial spraying by vaping happening at the same time (if vaping is allowed), especially considering that it involves a lesser contribution to a possible contagion than speaking, coughing or even continuous breathing for extended periods without mask wearing. While vaping would seem to represent a higher contagion risk because it is visible (while droplet emissions from speaking and cough are invisible), it is precisely the opposite: its visibility is what makes it safer because it clearly delineates its area of direct exposure (as shown in Figure 1).*Open outdoor spaces***.** For example, walking in the street or a park, or in a large volume covered space, like a stadium or a mall (assuming that vaping is allowed). Contagion risks significantly decrease with respect to those of indoor spaces, as droplet emissions are rapidly scattered and dispersed by the surrounding circulating air. Vapers in this scenario have much more incentives to wear face masks while not vaping (which involves 10–15 breaths per hour). Also, it is far easier in large open spaces for bystanders wearing masks to keep a recommended safe distance and to avoid the visible range of direct exposure delineated by the exhaled jet.

We remark that keeping a reasonable separation distance is as important for preventing contagion as face mask wearing, as the most commonly used masks (surgical or cotton) are far from achieving full efficiency in blocking direct exposure to emitted droplets. Mask usage for extended periods can be extenuating and cannot be rigidly maintained and enforced 100% of time in shared spaces, thus tolerating a margin of extra exposure due to intermittent face mask removal or adjustment is unavoidable and even necessary for civilized coexistence.

It is important to mention that vaping also involves contagion risks not related to aerial transmission, as it necessarily requires touching and tampering with a device inserted in the mouth and there is evidence of viral transmission through surfaces and fomites [49,50]. Vaping could also involve fomite contamination when touching and manipulating the mask to remove it in order to vape. However, the same risks of transmission through fomites are present in everyday activities, especially while drinking or eating, and even when manipulating the mask out of fatigue and discomfort for prolonged wearing without participating in any specific activity. These risks are unavoidable and can be easily tackled by simple hygiene prevention. There is no rational reason to emphasize these risks and to assign to them special concern only when vaping or smoking are involved.

Considering the points raised above, to prohibit vaping in fully open outdoor spaces alluding mask removal or possible fomite transmission has a weak and extremely speculative justification, more so in open spaces like restaurant terraces or outdoors. Unfortunately, the Inter-Territorial Council of the National Health System in Spain has precisely invoked in its positioning document [19] the need for protection of the public from COVID-19 contagion on such weak basis to justify a nationwide ban on smoking and vaping in all outdoor spaces (even fully open spaces) where an interpersonal separation distance of 2 m cannot be guaranteed, an intervention that is evidently unenforceable and subject to potential abuse and conflict.

Spanish health authorities do not provide empiric evidence that actual COVID-19 contagion through vaping or smoking exhalations has occurred nor a coherent technical justification supporting its plausibility, but nevertheless they invoke the precautionary principle to justify the enforcement of this ban at least for the duration of the pandemic. As we demonstrate in this paper on the basis of our rigorous analysis in [11] and further arguments described in Section 5 and Section 7, the visibility of the exhaled jet allows bystanders wearing face masks to avoid the risk for direct exposure (identified as the main contagion route), with the same masks protecting them from indirect exposure. To target exhalations from vaping and smoking as especially dangerous contagion vectors for COVID-19 is an extreme and invasive implementation of the precautionary principle that lacks a proper scientific basis derived from current knowledge on droplet dynamics and emissions of expiratory activities carrying the SARS-CoV-2 virus (or any other pathogen).

## 8. Conclusions

Since a significant number of vapers have quit smoking by taking up vaping, it is crucial that mitigation and prevention strategies do not lead to an environment (with or without confinement) that may cause these ex-smokers to relapse to smoking. This could happen if they (and/or their family members) become misinformed by exaggerated or misleading claims about vaping, like unsubstantiated claims that link vaping and COVID-19, or using the crisis of lung injuries that occurred in the USA in 2019 (the so-called “EVALI” or “e-cigarette, vaping lung injury” crisis) to issue the false claim that it is equally (or more) harmful than smoking. Vapers may also relapse to smoking if vaping shops are ordered to close while cigarettes remain accessible in convenience stores, as has happened in many jurisdictions [72,73,74] but not in others [75]. As commented before, cravings and anxiety can be undesired psychological byproducts of long term confinement and can increase consumption among smokers and vapers, or induce ex-smokers or ex-vapers to relapse. In this case, it is preferable to favor the increase of consumption or the relapse to e-cigarettes, the less harmful product.

The risk for direct and indirect COVID-19 contagion from indoor vaping expirations does exist and must be taken into consideration. However, this risk must be understood with reference to its potential to transport respiratory droplets in the context of markers and parameters of other expiratory activities. Therefore, as far as protection against the SARS-CoV-2 virus is concerned, vaping does not require particular additional interventions, other than those already suggested for the general public, in the home scenario or in shared social spaces: social distance and face masks. Vapers should be advised to be alert to non-vapers’ issues and worries while sharing indoor spaces or dwellings or when near to other residents, to use low-powered devices for low-intensity vaping for increasing safety and to keep a high standard of hygiene when using their devices. Vapers, however, also deserve sensitivity, courtesy and tolerance.

## Figures and Tables

**Figure 1 ijerph-18-01437-f001:**
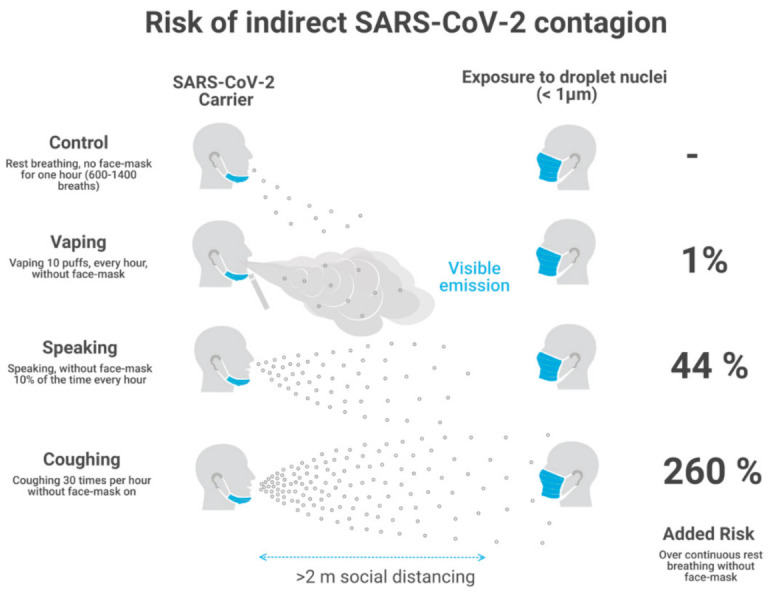
Direct and indirect exposure to various respiratory activities. The figure displays a sketch of the area of direct exposure for droplet emitters not wearing a face mask and masked bystanders. Notice that the flow of droplet emission is visible only for vaping, making it easy for bystanders to avoid direct exposure to it. The percentages in the right hand side denote risks of indirect exposure with respect to the control state of continuous breathing.

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
