# Peer review of "Aerial Transmission of the SARS-CoV-2 Virus through Environmental E-Cigarette Aerosols: Implications for Public Policies"

_ijerph, 2021, doi:10.3390/ijerph18041437_

Round 1

Reviewer 1 Report

  What do you want to do ? New mailCopy

Manuscript ID: ijerph-1065069

Title: Aerial transmission of the SARS-CoV-2 virus through environmental

e-cigarette aerosol: implications for public policies

Journal: International Journal of Environmental Research and Public Health

General comment:

This paper provides a viewpoint focusing on Aerial transmission of the SARS-CoV-2 virus through environ-2 mental e-cigarette aerosol: implications for public policies. This commentary appears quite interesting. However, the manuscript cannot be yet published since some major points should be deeply discussed.

Major Comment:

From the reviewer's point of view, this paper presents a major methodological problem. In the context of the COVID-19 pandemic, the face mask has become a major prevention tool to limit the spread of the disease. The authors have chosen to compare the risk of spread of SARS-CoV2 by:

- a vaper

- versus an individual breathing without a mask

In this specific case, I fully agree with the conclusions of the authors. This is particularly the case that can be encountered in the intra-family environment at home.

Nevertheless, this point of view is biased and does not seem to the reviewer to be the most relevant in terms of public health for cases of spraying "in public", "between friends", or "outdoor vaping" for which the wearing of a mask is recommended for all individuals. Indeed, in this case, the appropriate comparison is to evaluate the increase in risk between:

- a vaper

- versus an individual who should be masked!

In this case (which is perhaps the most critical for the management of the spread of COVID-19?) it is undeniable that removing one's mask in order to spray ECA will induce a significant increase in risk, which could even be critical in terms of contamination.

The authors very quickly address this case at the end of their manuscript in the section “Face Mask”. The authors say that ”We did not assume universal wearing of face masks”. I am sorry but I do not approve this strong assumption proposed by the authors. Besides, the authors also say that “In the analysis of the previous subsection, but even at this lowest level of prevention and protection, the analysis of [18] shows that vaping adds only a minuscule additional risk (1%) to those risks already existing from continuous breathing.” I am sorry but I do not understand the explanations proposed by the authors. The problem is precisely that in the case of significant protection against airborne spread of SARS-CoV2 (i.e. the case of an individual wearing a mask), vaping becomes an important risk factor for the airborne spread of the virus (perhaps no more important than the fact of breathing without a mask! we agree!). Finally, the authors say: “It can be argued that vaping requires the temporary removal of the vaper’s face mask and thus will raise the relative risk of exposure because of the decrease of the baseline level when everyone is wearing a face mask. However, this increase of risk would be inconsequential because it would be offset by the fact that the same face masks worn by everybody else would shield them from droplet emissions produced in the short intermittent mask-free vaping episodes.” How is it possible to state that "this increase in risk would be inconsequential because it would be offset by the fact that the same face masks worn by all the others would protect them from the droplet emissions produced during brief intermittent episodes of evaporation without masks"? Not only is this statement purely speculative, but the protection provided by face masks is only important if it is reciprocal! We should also take into consideration the protection of the vapers (without a mask) from the surrounding!

In my view, the conclusion of this comment should be: vaper does not bring a very significant increase in the risk of spreading and contaminating third parties compared to an individual breathing without a mask. However, since vaping requires not wearing a mask, and since not wearing a mask is an important factor in the spread of the virus by airborne means even over short periods of time, vaping is therefore a factor in the potential spread of SARS-COV2 at a level at least equivalent to not wearing a mask.

Minor Comments:

  1. “To claim that these findings provide strong evidence that vaping is a risk factor for COVID-19 is inconsistent with the noticeable absence of vapers among hospitalized or seriously ill COVID-19 patients. “

This statement is speculative, please give some literature reference.

  1. “In fact, aerosols with submicron droplets (like ECA) approximately evolve like gases with its particles behaving as molecular contaminants and are thus widely used to visualize respiratory flows”

from a physical point of view this assertion is false, this sentence is more of the popularisation of science which is not very relevant in a scientific publication

  1. “Since vaping involves mouth inhalation by suction through a mouthpiece, we review the available literature on the effects of the inspiration/expiration routes and of mouthpieces and nose clips on respiratory mechanics. “

A potential point of contamination by the electronic cigarette that has not been addressed by the proposed comment is the transmission of SARS-CoV2 through surface contact. When an object is worn nearly 200 times a day, which is very often handled and not decontaminated before each puff, the risk of contamination of electronic cigarette users through poor hand hygiene should not be negligible.

  1. “produce low emission rates: 2-230 respiratory droplets per puff, overwhelmingly in the submicron range (hence, they should be really droplet nuclei as droplets of this size evaporate instantaneously once exhaled). “

Reviewer 2 Report

The paper studies the implications for public policies of the aerial transmission of the virus SARS CoV2 through e-cigarettes smoking (or vaping), with reference to the contagion. For this reason I think that paragraph 2. Smoking and vaping as risk factor for COVID-19 is off topic and out of the scope declared in the title. In the study there is an extensive description of the emitted aerosol, but it lacks of mentioning the temperature, that can have a role in the aerosol generation and life and also an impact on the virus. There are many citations of a previous paper of the same authors (ref.18) but the link to this paper leads to a preprint. Was this paper published? and in this case are there redundant information? Paragraph 6.1 examines the risk during home confinement: living in the same home is a risk factor for contagion, independently from vaping or not. The authors affirm that vaping involves momentarily removal of the face masks: this is clearly not true, smokers, vapers, eaters and drinkers remove their mask from the face for long periods, significantly increasing the risk of contagion for bystanders. In conclusion the EVALI crisis is mentioned, the acronim should be explained.

Reviewer 3 Report

Dear Authors,

Your manuscript is really current and interesting.

It just needs some minor revision before accepting.

Why do You publish it as commentary? it could be an article too.

In introduction section just please better state the aim of the study, it should be extended.

In lines 169 Authors talk about surfaces and sars-cov-2 persistence, please update this with some recent literature:

1. Fiorillo, L.; Cervino, G.; Matarese, M.; D’Amico, C.; Surace, G.; Paduano, V.; Fiorillo, M.T.; Moschella, A.; La Bruna, A.; Romano, G.L., et al. COVID-19 Surface Persistence: A Recent Data Summary and Its Importance for Medical and Dental Settings. In Int J Environ Res Public Health, 2020; Vol. 17, p 3132.

2. Cervino, G.; Fiorillo, L.; Surace, G.; Paduano, V.; Fiorillo, M.T.; De Stefano, R.; Laudicella, R.; Baldari, S.; Gaeta, M.; Cicciù, M. SARS-CoV-2 Persistence: Data Summary up to Q2 2020. In Data, 2020; Vol. 5, p 81.

You should talk about different surfaces and temperature related persistence.

Kind Regards

Round 2

Reviewer 1 Report

/

Author Response

/

Reviewer 2 Report

The paper is largely improved.

However I still remind that a preprint cannot be cited in a paper.

 Sussman, R. A. Golberstein, E. and Polosa, R. Aerial transmission of SARS-CoV-2 virus (and pathogens in general) through 561
 environmental e-cigarette aerosol. medRxiv 2020.11.21.20235283;      doi: https://doi.org/10.1101/2020.11.21.20235283 562

Author Response

We respectfully disagree. Numerous pre-print citations are published every week by several scientific journals. The reviewer can verify this by himself/herself.

This manuscript is a resubmission of an earlier submission. The following is a list of the peer review reports and author responses from that submission.